# Criterion placement threatens the construct validity of neural measures of consciousness

**Johannes Jacobus Fahrenfort[1,2,3,4]\*, Philippa A Johnson[5], Niels A Kloosterman[6,7], Timo Stein[3,4], Simon van Gaal[3,4]**

[1]Department of Applied and Experimental Psychology, Vrije Universiteit Amsterdam, Amsterdam, Netherlands; [2]Institute for Brain and Behavior Amsterdam (iBBA), Vrije Universiteit Amsterdam, Amsterdam, Netherlands; [3]Department of Psychology, University of Amsterdam, Amsterdam, Netherlands; [4]Amsterdam Brain and Cognition, University of Amsterdam, Amsterdam, Netherlands; [5]Cognitive Psychology Unit, Institute of Psychology & Leiden Institute for Brain and Cognition, Leiden University, Leiden, Netherlands; [6]Department of Psychology, University of Lübeck, Lübeck, Germany; [7]Center of Brain, Behavior and Metabolism, University of Lübeck, Lübeck, Germany

**\*For correspondence:**
j.j.fahrenfort@vu.nl

## eLife Assessment

This **fundamental** study provides a critical challenge to a great many studies of the neural correlates of consciousness that were based on post hoc sorting of reported awareness experience. The evidence supporting this criticism is **compelling**, based on simulations and decoding analysis of EEG data. The results will be of interest not only to psychologists and neuroscientists but also to philosophers who work on addressing mind-body relationships.

**Abstract** How consciousness arises from brain activity has been a topic of intense scientific research for decades. But how does one identify the neural basis of something that is intrinsically personal and subjective? A hallmark approach has been to ask human observers to judge stimuli as 'seen' (conscious) and 'unseen' (unconscious) and use post hoc sorting of neural measurements based these judgments. Unfortunately, cognitive and response biases are known to strongly affect how observers place their criterion for judging stimuli as 'seen' versus 'unseen', thereby confounding neural measures of consciousness. Surprisingly however, the effect of conservative and liberal criterion placement on neural measures of unconscious and conscious processing has never been explicitly investigated. Here, we use simulations and electrophysiological brain measurements to show that conservative criterion placement has an unintuitive consequence: rather than selectively providing a cautious estimate of conscious processing, it inflates effect sizes in neural measures of both conscious and unconscious processing, while liberal criterion placement does the reverse. After showing this in simulation, we performed decoding analyses on two electroencephalography studies that employ common subjective indicators of conscious awareness, in which we experimentally manipulated the response criterion. The results confirm that the predicted confounding effects of criterion placement on neural measures of unconscious and conscious processing occur in empirical data, while further showing that the most widely used subjective scale, the Perceptual Awareness Scale (PAS), does not guard against criterion confounds. Follow-up simulations explicate how the experimental context determines whether the relative confounding effect of criterion placement is larger in neural measures of either conscious or unconscious processing. We conclude that

criterion placement threatens the construct validity of neural measures of conscious and unconscious processing.

## Introduction

Psychology has a long history of experimentally investigating the contents of the mind. After Watson rejected introspectionism (*Watson, 1914*), and the cognitive revolution in turn rejected behaviorism (*Baars, 1994*), it has become widely accepted that there is middle ground: one can potentially gain access to (some of) the contents of the mind by asking observers to report on these contents. This idea has been the central tenet in what has come to be known as the search for the neural correlate of consciousness (*Crick and Koch, 1990*; *LeDoux et al., 2020*). It was realized early on that to determine what consciousness is, one must contrast it with what is *not* conscious, or else the concept of consciousness is an empty shell (the contrastive approach, see *Baars, 1994*). This presupposes the idea that one can distinguish between conscious and unconscious processes (so-called dual process models). Indeed, the idea of a hidden unconscious life that precedes and/or escapes our conscious experience also has a long tradition, starting with the idea of unconscious inference (*Helmholtz, 1867*), and later in Sigmund Freud's hidden unconscious (*Freud, 1904*).

In the 80s and 90s of the previous century, this culminated in a heated debate centered around the question of how to experimentally establish whether a stimulus has reached consciousness or not. This debate roughly featured researchers defending the position that one should determine unconscious cognition at an objective detection threshold (*Greenwald, 1992*; *Greenwald et al., 1996*; *Greenwald et al., 1989*; *Snodgrass et al., 2004*), whereas others defended the position that only a subjectively defined threshold can establish unconscious cognition (*Cheesman and Merikle, 1986*; *Merikle, 1992*), with critical comments on the very notion of dual process models by questioning the existence of unconscious cognition at all (*Holender, 1986*; *Holender and Duscherer, 2004*).

The core difference between objective and subjective threshold models is that the subjective threshold approach claims to take consciousness seriously by letting the participants in a study indicate on their own terms whether they experience (see) a stimulus at a certain level of intensity (*Baars, 1994*). In contrast, the objective threshold approach attempts to establish the stimulus level at which some sensitivity measure is at chance, regardless of their claimed experience (Typically, participants have well above-chance objective sensitivity for presentation levels at which they claim that stimuli are 'subjectively' invisible *Stein et al., 2021*) and regardless of response criterion or bias (*Evans and Azzopardi, 2007*; *Balsdon and Azzopardi, 2015*; *Green and Swets, 1966*). Once it is established that some stimulus is either subjectively invisible (observer claims not seeing the stimulus) or objectively invisible (observer has zero sensitivity) at a given threshold, the typical recipe for establishing unconscious processing is to show that this stimulus still exerts behavioral effects (e.g. subliminal priming effects on a secondary task) or still undergoes residual neural processing (e.g. as measured through EEG or fMRI) despite being 'unconscious'.

Objective and subjective measures each have their own problems. Although objective measures seem to align best with a scientific approach to consciousness (replicable, objective, e.g. see *Mei et al., 2022*; *Soto et al., 2019*), they do require one to invoke a 'Gold Standard of seeing' that in fact does not exist (*Koenderink, 2014*). Relatedly, they ignore the fact that subjective experience is central to the very definition of consciousness. Indeed, cases have been reported in which subjective experience is reported to be different even when objective performance is equated (*Fleming et al., 2010a*; *Hesselmann et al., 2011*; *Lau and Passingham, 2006*; *Persaud et al., 2011*).

Furthermore, subjective measures are the measure of choice in paradigms in which physical stimulation is kept identical, which are often introduced to prevent that differences between conscious and unconscious vision can be attributed to physical rather than 'mental' differences (so-called 'threshold' approaches, *Sanchez et al., 2020*). These arguments question whether objective measures can even capture conscious experience, prompting many to defend subjective measures (*Baars, 1994*; *Dehaene, 2014*; *Overgaard et al., 2010*). Subjective measures on the other hand have been widely criticized for being confounded by effects that are unrelated to conscious experience, such as non-perceptual biases as well as regression to the mean effects (for critical reviews see *Newell and Shanks, 2014*; *Schmidt, 2015*; *Shanks, 2017*; *Soto et al., 2019*). Nevertheless, subjective measures have gained considerable popularity in consciousness research over the past 20 years (*Dehaene et al.,*

*2003*; *King et al., 2016*; *King and Dehaene, 2014*; *Lamy et al., 2009*; *Michel, 2022*; *Overgaard et al., 2010*; *Overgaard et al., 2006*; *Ramsøy and Overgaard, 2004*; *Salti et al., 2015*; *Sandberg et al., 2010*; *Sergent et al., 2005*; *Sergent and Dehaene, 2004*; *Soto et al., 2019*; *Soto et al., 2011*; *van Vugt et al., 2018*).

A dominant approach in the subjective threshold literature is to sort trials based on observer's responses to calculate the average neural activation for 'seen' (conscious) versus 'unseen' (unconscious) trials. Sorting of trials based on subjectively 'seen' or 'unseen' responses is known as post hoc sorting, because experimental conditions are established based on the participant's responses after the experiment has completed. Some proposed phenomena that originate from this approach are unconscious working memory (*King et al., 2016*; *Soto et al., 2011*; *Soto and Silvanto, 2014*; *Trübutschek et al., 2017*), unconscious error detection (*Charles et al., 2013*), and even unconscious arithmetic (*Sklar et al., 2012*), for critical comments see *Shanks, 2017*; *Stein et al., 2016*.

In this manuscript, we show that subjective measures are intrinsically prone to criterion confounds. Arbitrary criterion placement influences the decision about stimulus absence or presence, even when consciousness of the stimulus itself is not affected. Thus, two stimuli that undergo identical sensory processing and result in the same experience might either be reported as seen or as unseen, depending on whether the observer adopts a liberal versus a conservative criterion for deciding whether the threshold for a 'stimulus present' decision was reached. Although some may think that such criterion shifts must reflect changes in conscious experience, this is typically not the case. For example, it is well known that perceptual decisions may be motivated by non-perceptual information, such as the payoff matrix (the perceived utility of certain responses) or by statistical regularities in the environment, even when subjective experience is not affected (*Rungratsameetaweemana et al., 2018*; *Sánchez-Fuenzalida et al., 2023b*; *White and Poldrack, 2014*).

Indeed, when a large group of consciousness researchers at the Association of Scientific Studies of Consciousness (ASSC) conference was asked about the relationship between payoff-based criterion shifts and conscious perception, roughly two thirds answered that they did not think that such shifts involve changes in conscious perception (Q2 in *Francken et al., 2022*). The criterion problem has been known at least since the advent of signal detection theory (*Evans and Azzopardi, 2007*; *Green and Swets, 1966*) and has long been hypothesized to underly many – if not all – subjective threshold effects (*Eriksen, 1960*; *Goldiamond, 1958*; *Peters and Lau, 2015*; *Phillips, 2021*; *Phillips,*

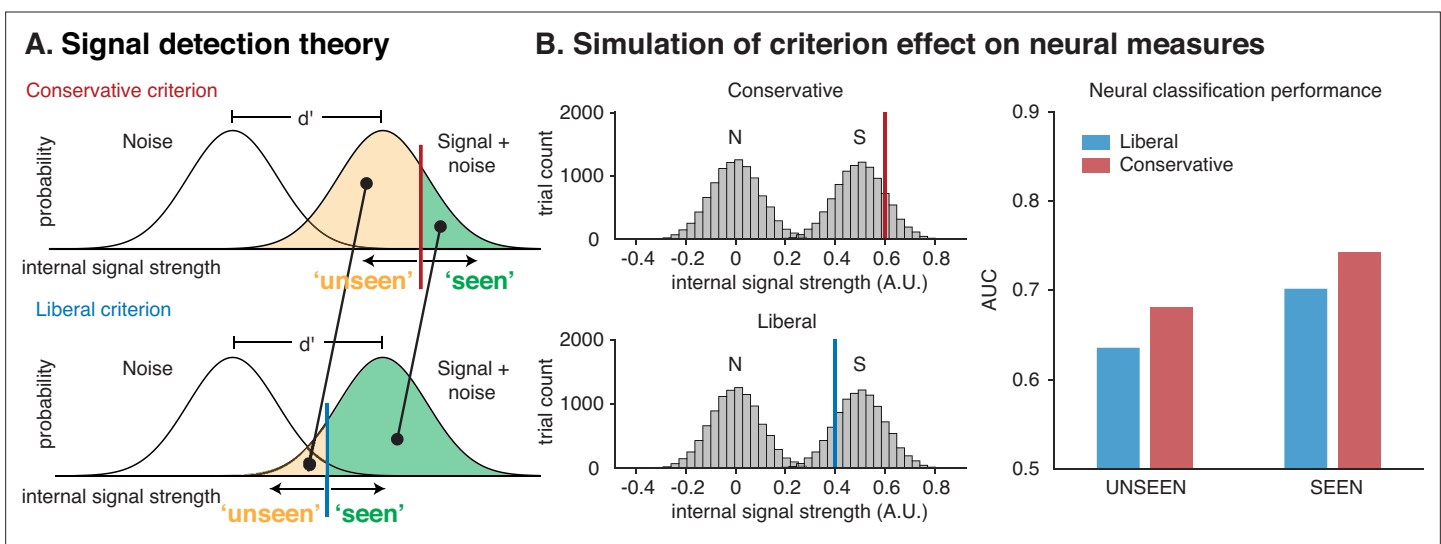

**Figure 1.** The effect of criterion shifts on neural measures of information processing under post hoc sorting. (**A**) When the criterion shifts to the right, the respondent becomes more conservative, whereas a leftward criterion shift reflects a more liberal response criterion. The average signal strength of both unseen and seen trials is greater under a conservative response criterion than under a liberal response criterion, both for seen and for unseen trials (diagonal line pointing leftward). (**B**) A simulation in which it is assumed that internal signal strength is a reflection of neural processing. When simulating signal and noise trials (histograms of signal and noise trials, left panel), a selection based on a conservative or liberal criterion would result in different average signal strengths, which would affect the sensitivity of a decoding analysis on neural data of signal versus noise for both seen and unseen trials, depicted as area under the curve (AUC) in the right panel.

*2016*). However, when criterion shifts are combined with post hoc sorting on subjective measures of consciousness, this may lead to further unwanted confounds. Here, we investigate the influence of criterion shifts on post hoc sorted neural measures of consciousness.

## Results

First, we sought to make explicit how – under the assumption of a signal detection theoretic model – neural measures of information processing are expected to behave when selecting trials based on a behavioral response. Interestingly, this exercise has never been undertaken. In signal detection theory (*Green and Swets, 1966*), the relationship between the response (behavior) of an observer and the signal that the observer operates on, depends on the criterion that the observer applies to that signal. This is depicted in *Figure 1A*, where a distribution of internal signals resulting from pure noise (distribution on the left), needs to be distinguished from a distribution that results when a signal is added to that noise (distribution on the right). How well the observer can distinguish signal from noise is determined by the distance between the two distributions (d', expressed in standard deviations).

To decide whether any given internal signal comes from the noise distribution or from the signal distribution, the observer must arbitrarily place a criterion somewhere (i.e. the threshold for responding in a certain way: red line/conservative in the top panel, blue line/liberal in the bottom panel). Signals strengths to the right of the criterion are then classified as 'seen', whereas signal strengths to the left are classified as 'unseen' by the observer. The subjective measures approach assumes that there is a 1:1 relationship between conscious perception and this criterion, but this need not be the case. Indeed, when levels of uncertainty are high, an identical internal signal may either result in a 'seen' or in an 'unseen' response depending on where this criterion was placed. Placement of the criterion can depend on many non-perceptual factors, including the perceived utility of certain stimulus–response combinations (the payoff matrix), statistical regularities in the environment (differences in the ratio of noise and signal presentations) but also on the state of the observer (explorative vs apprehensive) and even on small changes in task instructions.

When computing neural measures contingent on responses that are subject to criterion shifts, such uncontrolled criterion effects leak into neural measures that are based on behavioral response selection. For example, when the response criterion shifts from conservative to liberal, as is shown in the example in *Figure 1A*, the average signal strength within both the seen and the unseen response category decreases (see leftward diagonal lines from conservative to liberal). As a result, when extracting the signals from either response category, one should hypothetically get a decrease in the average signal strength in both 'unseen' and 'seen' conditions under a liberal criterion when compared to a criterion that is more conservative.

In *Figure 1B* we used a simple simulation to uncover the effect of criterion shifts on neural processing measures. In this simulation, we randomly generated normally distributed internal 'signals', which one may conceptualize as trials in a neuroimaging experiment in which a stimulus is presented. Next, we either applied a liberal or a conservative criterion to the signal strength histograms (*Figure 1B*, left panels) and computed the average signal strength of the 'unseen' (left of criterion) and 'seen' (right of criterion) conditions, separately for liberal and conservative (see Methods for details).

These average signal strengths are the equivalent of what would be termed neural measures of unconscious (unseen) and conscious (seen) processing in an experiment that uses subjective measures to establish experimental conditions. Somewhat counterintuitively, a more conservative criterion does not have different effects on the neural measure of 'unconscious' processing and on the neural measure of 'conscious' processing. Instead, when the two criteria are positioned symmetrically around the mean of the signal distribution as in the left panels of *Figure 1B*, a more conservative criterion inflates neural measures of both unconscious and conscious processing when compared to a liberal criterion (see the right panel of *Figure 1B*). Although this consequence of criterion shifts on effect sizes in neural measures based on post hoc sorted trials is clearly implied by signal detection theory, to our knowledge it has not been highlighted in the consciousness literature.

Next, we wondered to what extent such criterion effects become apparent in neural measures of unconscious and/or conscious processing when this method is applied to empirical data. To investigate this, we analyzed two datasets in which a criterion manipulation was applied to a detection task. In both experiments, participants viewed a continuous rapid serial visual presentation (RSVP) of oriented textures while EEG was collected (see *Figure 2A, B*). The sequence of textures was always

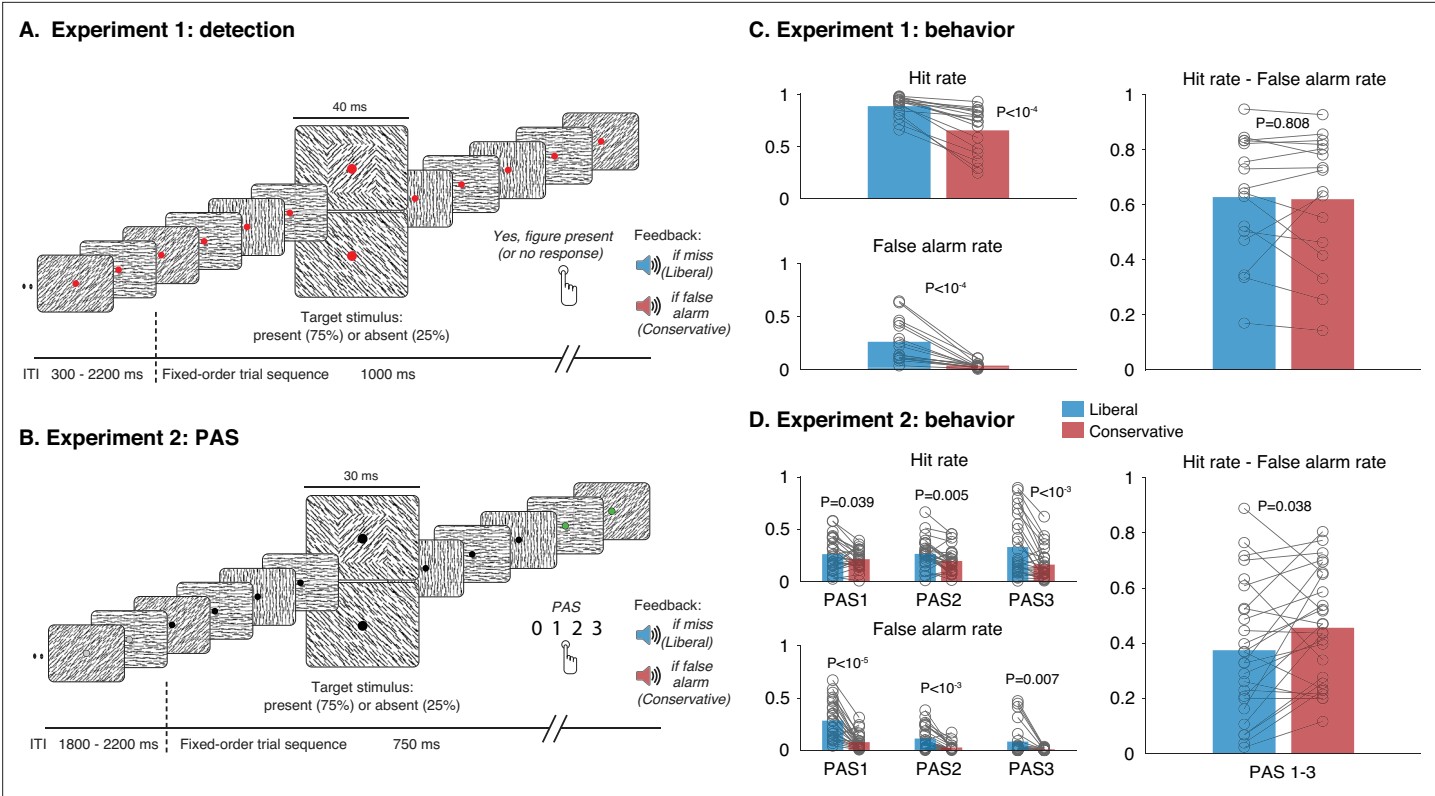

**Figure 2.** Two experiments with a criterion manipulation. (**A**) In Experiment 1, participants had to execute a detection task in 10-min blocks by pressing a button whenever they perceived a square target (top stimulus in a continuous rapid serial visual presentation [RSVP]). In half of the blocks, misses were signaled by a tone and a small monetary deduction (liberal criterion condition) whereas in the other half of the blocks false alarms were signaled by a tone and a small monetary deduction (conservative criterion condition). (**B**) Same as in panel A, but now for Experiment 2, in which participants responded according to the PAS. (**C, D**) Behavior associated with Experiments 1 and 2. Both hits and false alarms increased under the liberal condition compared to the conservative condition, whereas sensitivity remained approximately the same. Note that in panel D, to be able to compute hit and false alarm rates the PAS was conceptualized as a type I response scale, with PAS0 indexing absent responses and PAS1–3 indexing present responses. Minor differences between experiments are detailed in the Methods.

identical, except that the fifth texture either contained a texture-defined square of lines orthogonal to the background orientation (target trials) or a homogenous texture without such a square (no-target trials). In Experiment 1, which has previously been published by *Kloosterman et al., 2019*; *Kloosterman et al., 2020*, the participant's task was to identify the target by pressing 'yes' whenever they observed a square (detection experiment, see *Figure 2A*). To manipulate their decision criterion, they were either punished using an aversive tone with a small monetary deduction for misses (liberal condition) or an aversive tone with a small monetary deduction for false alarms (conservative condition).

In a second experiment, we had a different group of participants perform the same task, this time responding using the Perceptual Awareness Scale (PAS: *Ramsøy and Overgaard, 2004*). This scale allows observers to indicate the strength of their experience at a more fine-grained resolution ranging from [0] 'No experience' to [3] 'A clear experience' (PAS experiment, see *Figure 2B* and Methods for full description of the four response categories). The underlying assumption of the PAS is that selecting [0] will only occur if trials are 'truly' unseen, so that unconscious processing is not overestimated, as may happen in dichotomous or other types of scales (*Overgaard et al., 2006*; *Overgaard and Sandberg, 2021*; *Sandberg et al., 2010*). The PAS was developed to be able to resolve the inability to externally calibrate subjective content, and as such its ultimate goal seems to be to be impervious to non-perceptual criterion shifts. However, despite its popularity, the degree to which the PAS is robust to non-perceptual criterion shifts has never been explicitly investigated. Here, to counter criterion shifts when using the PAS, we further explicitly instructed observers to respond in line with their experience: '*Even though you receive feedback about the correctness of your responses, it is very important that you keep responding according to what you actually experience, using the Perceptual*

*Awareness Scale. Only press 0 if you are 100% convinced that no square appeared and only press 3 if you are 100% convinced that a square appeared.*' The criterion manipulation was applied by counting PAS0 responses on target trials as misses, and PAS1, PAS2, or PAS3 responses on no-target trials as false alarms (as is common practice, e.g. *Soto et al., 2011*). Aside from response mode and target presentation duration (Experiment 1: 40 ms, Experiment 2: 30 ms), both experiments were virtually identical (see Methods for other minor differences).

Behaviorally, the criterion manipulation (liberal vs conservative) resulted in a strong criterion shift in both experiments, expressed in concomitant increases of both hits (responding 'seen' when a target was presented) and false alarms (responding 'seen' when no target was presented) for liberal when compared to the conservative condition (see left panels of *Figure 2C, D*). In contrast, sensitivity – simplified here as the hit rate minus the false alarm rate – remained largely unchanged (right panel of *Figure 2C, D*). To quantify the success of the criterion manipulation in Experiment 1, we computed the signal theoretic parameter estimates criterion c (liberal: –0.30, conservative: 0.73) and sensitivity d' (liberal: 2.12, conservative: 2.39). This confirmed that the criterion manipulation in Experiment 1 was successful, and that it exerted a much larger effect on criterion (hedges $g$ = 2.84) than on d' (hedges $g$ = 0.30).

The behavioral data of Experiment 2 were analyzed in the same way as Experiment 1, this time sorting trials using the four PAS levels, conceptualizing the PAS as a type I response scale, with PAS0 indexing absent responses and PAS1–3 indexing present responses. We reasoned that if participants can maintain a stable response criterion reflecting their experience under the PAS, we should not observe criterion shifts. In contrast, however, we observed strong criterion shifts on all levels of the PAS, as can be seen from concomitant increases of both hits and false alarms for liberal when compared to the conservative condition (see left panels of *Figure 2D*), while sensitivity (collapsing PAS1, PAS2, or PAS3 as 'seen' responses) was only slightly higher for conservative compared to liberal (see the right panel of *Figure 2D*). This shows for the first time, that all levels of the PAS are affected by criterion shifts, questioning the construct validity of the measure (see Discussion). Computing the corresponding signal theoretic estimates for Experiment 2 confirms this result for criterion c (liberal: –0.59, conservative: 0.60) and sensitivity d' (liberal: 1.29, conservative: 1.59). Effect sizes on these measures show that the criterion manipulation in Experiment 2 worked and that it exerted a much larger effect on criterion (hedges $g$ = 2.02) than on d' (hedges $g$ = 0.42), just as we observed in Experiment 1.

Next, to establish the effect of criterion shifts on neural measures, we turned to classification performance of EEG data as a measure of neural processing (*Fahrenfort et al., 2018*). First, a linear discriminant analytic (LDA) classifier was trained for each participant using all trials from all sessions (three sessions in Experiment 1, two sessions in Experiment 2) to discriminate target from no-target trials based on EEG data, irrespective of seen/unseen responses and irrespective of the response criterion. To maximize signal-to-noise ratio, we applied a leave-one-person-out cross validated decoding scheme by using all classifiers from all participants except the participants that was being tested (separately for Experiment 1 and for Experiment 2). This leave-one-person-out cross validation procedure maximized the available data for training without requiring k-folding on subsets of cells with low response counts, so that all test sets were classified by the same fully independent classifiers. A single time series of classification performance across time was obtained for every participant (every testing set) by averaging classification performance across all classifiers that tested that set (see Methods and *Figure 3—figure supplement 2* for details). We maximized signal-to-noise ratio by performing classification using occipitoparietal electrodes, as these are known to be most sensitive to these stimuli (*Fahrenfort et al., 2017*; *Fahrenfort et al., 2008*; *Fahrenfort et al., 2007*). Different electrode selections (all electrodes or only occipital electrodes) yielded qualitatively similar results.

In a first step, we computed classifier performance over time across all trials (irrespective of responses or condition) in the experiment, separately for Experiments 1 and 2. We subsequently computed the average classification performance across both experiments (see *Figure 3A*, left panel) and identified three times at which local maxima occurred in this average (137, 266, and 430 ms). These peaks reflect stages that are often identified in similar experiments that investigate the time course of perceptual organization (*Fahrenfort et al., 2017*; *Fahrenfort et al., 2008*; *Fahrenfort et al., 2007*). *Figure 3A* (right panel) shows that these peaks have highly similar topographic maps of current source density, obtained from the forward transformed weights of the training data from

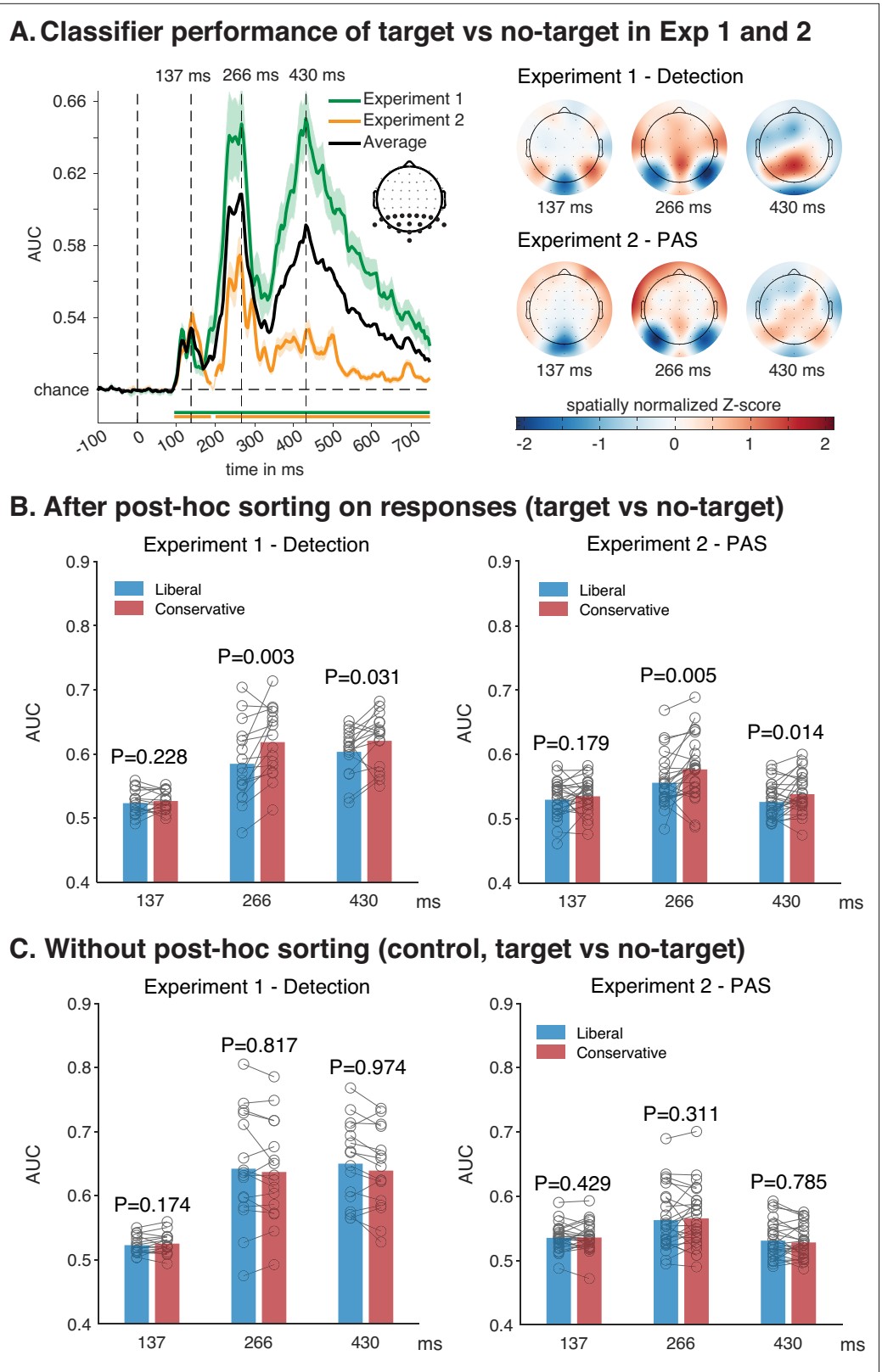

**Figure 3.** Classification performance with and without post hoc sorting. (**A**) Classification performance of target present versus target absent time-locked to stimulus onset, expressed in area under the curve (AUC), separately for Experiment 1 (N=16) and Experiment 2 (N=26), as well as average classification performance, shaded areas are standard error of the mean (left panel). The average performance across both experiments clearly shows

*Figure 3 continued on next page*

*Figure 3 continued*

three local classification performance peaks at 137, 266, and 430 ms. The distribution of cortical activity at these peaks was highly similar for Experiments 1 and 2, as shown in topographic current source density maps that were obtained from the forward transformed classification weights, obtained from training a classifier using all electrodes for visualization purposes (right panel). (**B**) Classification performance of target present versus target absent after post hoc sorting on 'seen' and 'unseen' responses (here collapsed across 'seen' and 'unseen', see *Figure 4* for the uncollapsed data), separately for Experiments 1 and 2, and separately for the liberal and the conservative condition. (**C**) Same as in B, but this time performing classification analysis on all trials without first post hoc sorting into 'seen' and 'unseen' trials (using the same classifiers as used in B). See *Figure 3—figure supplement 1* for the complete time series.

The online version of this article includes the following figure supplement(s) for figure 3:

**Figure supplement 1.** Decoding timelines of liberal and conservative with post hoc sorting (**A**) and without post hoc sorting, control analysis (**B**).

**Figure supplement 2.** Graphical depiction of leave-one-person-out cross validation scheme.

the selected peaks for both experiments (*Haufe et al., 2014*). To keep multiple comparisons to a minimum, further testing was carried out on the time points at which these three peaks occurred. Note that classification performance was higher for Experiment 1 than for Experiment 2, as this experiment had 3 sessions instead of 2, and target stimuli were presented for 40 ms instead of 30 ms (which also resulted in a higher behavioral d' for Experiment 1 than for Experiment 2). Note that this has no bearing on the relevant hypothesis tests, because hypothesis testing effects are only within, and not between experiments (i.e. the only relevant difference is between conservative and liberal, not between Experiments 1 and 2).

Next, to investigate the effect of these criterion shifts on neural measures, we determined how classification performance was affected by post hoc sorting trials based on response categories, separately for decisions made under a liberal criterion and under a conservative criterion. We applied a standard post hoc sorting procedure to the EEG data of each experiment, creating a conscious condition of 'seen' figure trials and an unconscious condition of 'unseen' figure trials, separately under a conservative and a liberal criterion. For Experiment 1, the difference between seen and unseen was operationalized as participants either giving a 'yes' response (='seen') or no response at all (='unseen'). For Experiment 2 the difference between seen and unseen was operationalized as giving a response of 1, 2, or 3 on the PAS (='seen') or giving a 0-response on the PAS (='unseen').

To ensure that differences resulting from post hoc sorting could not be explained by differences in signal-to-noise ratio resulting from disparities in trial counts in the testing set, we equated trial counts between the liberal and conservative condition within each participant by randomly selecting the same number of trials from overrepresented cells (for Experiment 1, this was done at the level of 'seen' and 'unseen' responses, for Experiment 2 the trial counts were equated at each of the PAS levels, see methods for details). As a result, response-contingent conditions in the liberal and conservative conditions had identical input for all classification analyses. Although different trial counts in the testing set might affect the precision with which area under the curve (AUC) is estimated in a decoding analysis, it does not affect the size of AUC itself. Trial count equation was merely performed to make sure the liberal and conservative condition were as comparable as possible. Analyzing the data without equating trial counts resulted in qualitatively identical results.

We then extracted classifier performances at the aforementioned peaks under both a conservative and a liberal criterion, separately for the 'seen' and 'unseen' conditions, and separately for Experiments 1 and 2. For Experiment 2, we initially collapsed (averaged) the three 'seen' PAS levels (1, 2, and 3) into a single 'seen' level, so that the visibility factor of Experiments 1 and 2 would have the same two levels. We then entered these in a large 2 (experiment) × 2 (visibility) × 2 (criterion) × 3 (latency) repeated measures ANOVA with experiment as a between group factor. The result of this ANOVA showed strong main effects of experiment (Exp1 vs Exp2: $F_{1,40} = 38.68$, $p < 10^{-6}$, $\eta^2_p = 0.49$), visibility (seen vs unseen: $F_{1,40} = 204.01$, $p < 10^{-16}$, $\eta^2_p = 0.84$), criterion (liberal vs conservative: $F_{1,40} = 20.98$, $p < 10^{-4}$, $\eta^2_p = 0.34$), and latency (137, 266, and 430 ms: $F_{1.79,80} = 33.72$, $p < 10^{-8}$, $\eta^2_p = 0.46$), see *Supplementary file 1a* for the full ANOVA.

First, we asked whether the criterion shift significantly affected classification performance after post hoc sorting, which was confirmed by the highly significant main effect of criterion. This can also

be seen in *Figure 3B*, where we first show the main effect of criterion separately for Experiment 1 (detection) and Experiment 2 (PAS) collapsed across 'seen' and 'unseen' trials. These data confirm – as was predicted from our simulation – that post hoc sorting results in a large criterion effect, with higher classification performance for the conservative than for the liberal condition. The consistency of the effect is further supported by the fact that criterion did not significantly interact with experiment ($F_{1,40}$ = 1.38, p = 0.25, $\eta^2_p$ = 0.03). Criterion did interact with visibility ($F_{1,40}$ = 5.36, p = 0.026, $\eta^2_p$ = 0.12), and with latency ($F_{1.75,69.80}$ = 4.44, p = 0.02, $\eta^2_p$ = 0.1), indicating that criterion effects on neural measures manifested differentially in different visibility levels and at different moments in time. Indeed, in both experiments the criterion effect occurs at long latencies (266 and 430 ms) but not at the short latency of 137 ms, as was established using one-sided *t*-tests (conservative >liberal) for simple effects in each of the three latencies for each of the two experiments (see *Figure 3B*, for the complete time courses of these experiments see *Figure 3—figure supplement 1*). Finally, there was a highly significant three-way interaction between criterion, experiment, and visibility ($F_{1,40}$ = 18.47, p < $10^{-3}$, $\eta^2_p$ = 0.32), which we will expound on in much more detail further down below.

An alternative explanation for these findings might be that the criterion effect is not driven by response-contingent post hoc sorting, but rather that the conservative condition just has higher classification accuracy overall (regardless of post hoc sorting). To investigate this possibility, we re-analyzed the same data without post hoc sorting, i.e. by taking all trials in the conservative condition and in the liberal condition without sorting them into 'seen' and 'unseen'. As in the previous analysis (and using the same classifiers), we extracted classification performance at these peaks under a conservative and under a liberal criterion, separately for Experiments 1 and 2 (see *Figure 3C*, for the complete time courses see *Figure 3—figure supplement 1B*).

To test for criterion effects in the absence of post hoc sorting, we then entered these data in a 2 (experiment) × 2 (criterion) × 3 (latency) repeated measures ANOVA with experiment as a between group factor (naturally there was no factor visibility because there was no post hoc sorting). As before, the result of this ANOVA showed strong main effects of experiment (Exp1 vs Exp2: $F_{1,40}$ = 28.45, p < $10^{-5}$, $\eta^2_p$=0.42), and latency (137, 266, and 430 ms: $F_{1.70,67.95}$ = 49.95, p < $10^{-12}$, $\eta^2_p$ = 0.56), but this time there was no main effect of criterion (liberal vs conservative: $F_{1,40}$ = 1.16, p = 0.29, $\eta^2_p$ = 0.03), see *Supplementary file 1b* for the full ANOVA. *Figure 3C* shows the effect of the criterion manipulation separately for the two experiments, with one-sided post hoc *t*-tests (conservative > liberal) for each of the three latencies. This figure shows that – if anything – the criterion effect is in the opposite direction of the effect observed in *Figure 3B*, further confirming that the criterion effects in *Figure 3B* are due to post hoc sorting and not due to general effects of decoding sensitivity in the conservative versus the liberal condition.

Having established the specificity of the post hoc sorting effect, we return our attention to the initial post hoc sorting analysis (*Figure 3B* and *Supplementary file 1a*). In this analysis, we observed a small two-way interaction between visibility and criterion as well as a highly significant three-way interaction between experiment, criterion, and visibility as noted above ($F_{1,40}$ = 18.47, p < $10^{-3}$, $\eta^2_p$ = 0.32), also see *Supplementary file 1a*. This would suggest that Experiments 1 and 2 contain different post hoc sorting criterion effects on the 'seen' and the 'unseen' condition. To investigate this further, we performed separate ANOVAs for Experiment 1 (*Figure 4A*) and Experiment 2 (*Figure 4B*).

For Experiment 1, we performed a 2 (visibility) × 2 (criterion) × 3 (latency) repeated measures ANOVA (see *Supplementary file 1c* for the full ANOVA). This analysis again shows strong main effects of visibility ($F_{1,15}$ = 81.38, p < $10^{-6}$, $\eta^2_p$ = 0.84), criterion ($F_{1,15}$ = 13.89, p < 0.01, $\eta^2_p$ = 0.48), and latency ($F_{2,30}$ = 37.13, p < $10^{-8}$, $\eta^2_p$ = 0.71), but more importantly, it also shows that criterion strongly interacts with visibility in Experiment 1 ($F_{1,15}$ = 11.06, p < 0.01, $\eta^2_p$ = 0.42). One-sided *t*-tests for simple effects (conservative > liberal) revealed that the criterion effects were only significant in the 'unseen' condition (*Figure 4A*, left panel), but not in the 'seen' condition (*Figure 4A*, right panel), and – as established before – that these effects only appear in the late 266/430 ms latencies as opposed to the early 137 ms latency.

To investigate the interaction between visibility and criterion in Experiment 2, we performed the same analysis in a 4 (visibility) × 2 (criterion) × 3 (latency) repeated measures ANOVA, this time maintaining the four PAS responses as separate levels in the factor visibility (see *Supplementary file 1d* for the full ANOVA). Again, this analysis confirmed strong main effects of visibility ($F_{2.37,59.22}$ = 67.29, p < $10^{-16}$, $\eta^2_p$ = 0.73), criterion ($F_{1,25}$ = 10.10, p = 0.004, $\eta^2_p$ = 0.29), and latency ($F_{1.89,47.28}$ = 13.26, p < $10^{-4}$,

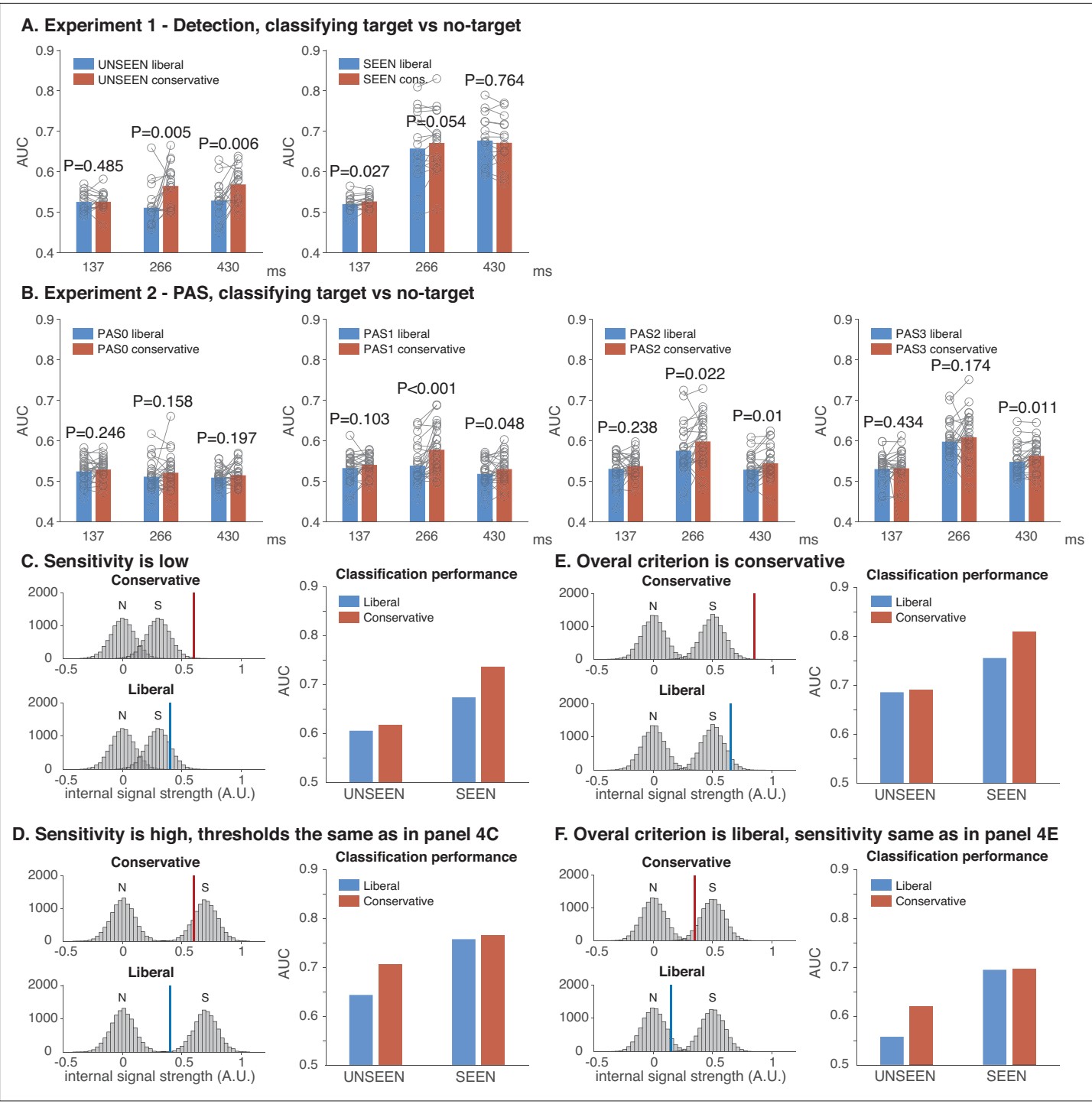

**Figure 4.** Differential effects on classification performance of post hoc sorting on either 'seen' or 'unseen' conditions. (**A**) Classification performance in Experiment 1 for the liberal and conservative condition, separately for 'unseen' trials (left panel) and for 'seen' trials (right panel). (**B**) Same as in A, but now for Experiment 2. (**C, D**) Simulations showing the effect of criterion shifts under low sensitivity or high sensitivity when the threshold is the same in C and D. When overall sensitivity is low (**C**), the effect of criterion shifts is more likely to appear in the 'seen' condition, whereas when sensitivity is high (**D**), the effect of criterion shifts is more likely to appear in the 'unseen' condition. (**E, F**) The effect of criterion shifts under an overall conservative or an overall liberal criterion when sensitivity is the same. When the overall criterion is conservative (**E**), the effect of criterion shifts is more likely to appear in the 'seen' condition, whereas when the overall criterion is liberal (**F**), the effect of criterion shifts is more likely to appear in the 'unseen' condition.

$\eta^2_p$ = 0.35), as observed in Experiment 1. Moreover, this experiment-specific ANOVA confirms that here too, criterion interacts with visibility ($F_{2.29,41.37}$ = 2.86, p = 0.06, $\eta^2_p$ = 0.10). However, the direction of the interaction is very different from Experiment 1. Post hoc t-tests for simple effects (conservative > liberal) revealed that in Experiment 2, the criterion effects were predominantly in the 'seen' conditions of the PAS (PAS1, PAS2, and PAS3, see *Figure 4B*, second, third, and fourth panels), but not in the 'unseen' condition (PAS0, see *Figure 4B*, leftmost panel), whereas Experiment 1 showed a significant effect in 'unseen' but not in 'seen' (and again only at late 266 and/or 430 ms latencies). These differential effects of the criterion on 'seen' and 'unseen' across experiments explains the highly significant three-way interaction between criterion, experiment, and visibility in the initial ANOVA in which experiment was incorporated as a factor.

Together, these results confirm clear criterion effects in both Experiments 1 and 2 due to post hoc sorting, as was predicted from our initial simulations. However, comparing Experiments 1 and 2 also shows that these criterion effects are most prominent in the 'unseen' condition for Experiment 1, while being most prominent for the 'seen' conditions (PAS1, PAS2, and PAS3) in Experiment 2. Wondering why this might be, we went back to our simulations to determine what parameters influence whether a criterion shift is expressed more strongly in 'unseen' or in 'seen' conditions. To determine this, we separately manipulated two main parameters in our model: sensitivity (the distance between the noise and the signal distribution) and *overall* criterion (whether the conservative and liberal criterion are either on the left or on the right side of the signal distribution).

*Figure 4C–F* shows the result of these simulations. First, we manipulated the degree of sensitivity (*Figure 4C, D*), while keeping the response thresholds for responding 'seen' or 'unseen' the same. When sensitivity is low (*Figure 4C*, left panels), a small criterion shift is more likely to have a large effect on the neural measure in the 'seen' condition after post hoc sorting (*Figure 4C*, right panel). Conversely, when sensitivity is high (*Figure 4D*, left panels) the same criterion shift is more likely to have a large effect on the 'unseen' condition (*Figure 4D*, right panel). These simulations show that under identical signal detection thresholds, a change in sensitivity (e.g. when showing higher strength stimuli, or when an observer is more attentive), can have a large differential impact on the neural measures in 'seen' and 'unseen' conditions, making their interpretation intrinsically problematic.

In a second simulation, we looked at the effect of overall criterion shifts (*Figure 4E, F*), while keeping sensitivity the same. When the overall criterion is conservative (*Figure 4E*, left panels), a small criterion shift might have a large effect on the neural measure in the 'seen' condition after post hoc sorting (*Figure 4E*, right panel), whereas when the overall criterion is liberal (*Figure 4F*, left panels) the same criterion shift can have a large effect in the 'unseen' condition (*Figure 4F*, right panel). This shows that small criterion shifts can have a differential impact on the effect size in neural measures of 'seen' and 'unseen' stimuli, even when sensitivity stays the same, depending on whether the overall criterion is liberal or conservative, again making the interpretation of such effects problematic.

Thus, criterion shifts can differentially confound neural measures of conscious or unconscious processing, depending both on overall criterion and sensitivity. The direction of this influence can be counterintuitive, is difficult to predict, and cannot be controlled experimentally. Interestingly, Experiment 1 has a much higher sensitivity than Experiment 2 (both in terms of classification accuracy and in terms of behavioral sensitivity, as pointed out before). This provides a compelling explanation for our finding that Experiment 1 (detection) shows the largest effect of the criterion shift in the unseen condition, whereas for Experiment 2 (PAS) the largest effect was observed in the PAS1 condition and up, which is nicely in line with the predictions from our simulation in *Figure 4C, D*. Importantly, these factors either cannot be controlled in principle (criterion) and/or are not controlled in practice (sensitivity), which questions the construct validity of subjective measures under post hoc sorting. In the discussion, we consider what these results mean for the future of subjective measures in consciousness research.

## Discussion

In this manuscript, we have shown through simulation that post hoc sorting on 'seen' and 'unseen' responses causes neural effect sizes in 'seen' and 'unseen' conditions to become stronger under a conservative compared to a liberal response criterion. To test this claim empirically, we analyzed data from two experiments in which criterion shifts were induced experimentally (one using a simple detection task, one using the PAS). We found that in both experiments, criterion shifts modulated

effect size in neural measures of 'unconscious' (unseen) and/or 'conscious' (seen) processing, and that this happens even though the conservative and liberal condition used the same independent training data (identical classifiers), and even though the trial counts in the test sets were equated for the conservative and liberal condition. Finally, we showed through simulation that such criterion shifts can either predominantly impact the neural measure of 'unconscious' or of 'conscious' processing (or both), depending on the sensitivity of observers and how they place their criterion based on the experimental context.

Together, these data show that criterion shifts confound neural measures of conscious and unconscious processing under post hoc sorting. Such confounds potentially contaminate studies of unconscious cognition (*King et al., 2016*; *Soto et al., 2011*) as well as studies that apply post hoc sorting to reveal neural measures of consciousness through a contrastive approach (*van Boxtel et al., 2010*; *Hesselmann et al., 2011*; *Hesselmann and Malach, 2011*; *Levinson et al., 2021*; *Melloni et al., 2011*; *Ress and Heeger, 2003*; *Rolke et al., 2001*; *Salti et al., 2015*; *Sanchez et al., 2020*; *Sergent et al., 2005*; *Stein et al., 2021*; *van Vugt et al., 2018*; *Wyart and Tallon-Baudry, 2008*). Concretely, the contrast 'seen > unseen', used in neuroimaging studies to isolate the neural basis of consciousness, can either result in strong or in weak differences between conditions, depending on the response criterion that was adopted by the observers.

Unfortunately, arbitrary criterion placement is intrinsic to decision making, and is known to be affected by many factors that are not controlled between, or even within experiments. Context is known to have a large effect on how observers place their criterion. For example, the ratio of targets to non-targets (the base-rate), the strength of the target stimulus compared to noise (the signal-to-noise ratio), the utility of certain stimulus–response combinations (the payoff matrix), and even small changes in task instructions are all known to affect the response criterion (*Fleming et al., 2010b*; *Kloosterman et al., 2019*; *Rakhshan et al., 2020*; *Supèr et al., 2001*; *White and Poldrack, 2014*). So even if some criterion shifts may have a perceptual origin (*Meyerhoff and Scholl, 2018*; *Witt et al., 2015*), the omnipresence of arbitrary criterion placement due to non-perceptual influences (*Sánchez-Fuenzalida et al., 2023a*; *Sánchez-Fuenzalida et al., 2023b*) threatens the construct validity of subjective measures in consciousness research.

Relatedly, criterion shifts also threaten the construct validity of the PAS. Experiment 2 shows that all levels of the PAS are sensitive to the criterion manipulation, even when explicitly instructing participants to only respond according to their experience. This finding is in line with evidence showing similar effects on related subjective measures, such as confidence judgements. For example, both payoff and base-rate induced criterion shifts not only result in a bias on first order decisions, but also affect how second order confidence judgements are distributed (*Lebreton et al., 2019*; *Lebreton et al., 2018*; *Locke et al., 2020*), also see *Peters et al., 2017*. Indeed, recent experiments from our lab have confirmed that payoff and base-rate induced criterion shifts affect confidence scores even when perceptual experience is not affected (*Sánchez-Fuenzalida et al., 2023a*).

Some may argue that the usage of the PAS in a context in which criterion shifts are experimentally induced is not the spirit in which the PAS was devised. One might even claim that Experiment 2 does not make use of the PAS because the criterion was explicitly manipulated in this study, and that one should always take care to extensively calibrate the PAS to subjective content in every experimental context (*Sandberg and Overgaard, 2015*). However, proper instruction does not resolve this issue. Any context – experimentally induced or not – influences the criterion in some way, and if there is no way to enforce that the adopted criterion is an unwavering threshold on subjective experience (and nothing else), any subjective measure is potentially prone to confounds of a non-experiental nature. Without resolving this, any experiment remains open to the critique that the participants in the study may not have adopted the subjective scale as it was intended, for whatever reason.

Indeed, critics of the current experiment would voice exactly this concern: that the participants did not adopt the PAS as it was intended due to wrong instruction and/or due to the experimental context. In particular, Sandberg and Overgaard have noted (personal communication) that one of the instructions we used (the sentence 'Only press 0 if you are 100% convinced that no square appeared and only press 3 if you are 100% convinced that a square appeared') as well as the use of feedback/payoff differs from the way PAS has been used in other studies (*Sandberg and Overgaard, 2015*). Future research would need to show whether removing these manipulations would (partially) mitigate the criterion effects on the PAS that we found in Experiment 2. Nevertheless,

criterion shifts are well known to also occur without these manipulations and occur naturally across experiments that have different ratios of target to non-targets, different overall levels of visibility and so forth.

As such, the current experiment can be viewed as a caricature of actual experimental practice. For example, depending on how 'calibration to subjective content' is done and the experimental context that is generated, some may show that blindsight does not exist (*Mazzi et al., 2016*; *Overgaard et al., 2008*), while others may use the same subjective scale to show that unconscious working memory does exist (*King et al., 2016*; *Soto et al., 2011*). Plausibly, such patterns of results can be reversed when authors would adopt different calibration procedures or invoke different experimental contexts in their experiments, without an objective way of maintaining or quantifying the construct validity of the adopted subjective scale. Importantly, there is no such thing as a criterion-free experimental context. As pointed out in the beginning of this discussion and in our simulations, many dimensions that are not controlled between experiments, will have a large effect on the criterion even without manipulating it explicitly/experimentally.

Thus, researchers studying consciousness are faced with a difficult conundrum. On the one hand, many view subjective measures as a crucial behavioral marker of the presence or absence of consciousness (*Baars, 1994*; *Dehaene, 2014*; *Overgaard et al., 2010*). On the other hand, subjective measures do not reliably measure the construct they intend to measure due to criterion confounds. A potential way out of this conundrum has been a proposal by *Peters and Lau, 2015*. They combined an objective, first order judgment with a criterion-free subjective measure by having participants place bets (as proxy for subjective reports) on their objective judgment. Trials consisted of two intervals, only one of them containing a grating target, and subjects were asked to guess grating orientation for both intervals and place a bet on which of these two judgments is deemed as more likely to be correct. They argued that above-chance objective performance in the absence of subjective insight measured in this criterion-free way would reflect unconscious perception. One may debate however whether this would save the intended nature of subjective measures, as the bet that participants are forced to place on one of the two intervals seems equivalent to an objective two-interval forced choice task. Without making any final judgment on this matter, we point out that saving the construct validity of subjective measures requires one to solve the criterion problem (*Morgan et al., 2013*). Without a properly experimentally defined procedure for doing so, their construct validity will remain under threat.

Summarizing, aside from the effect of the criterion on behavioral responses themselves, we show both empirically and in simulation that post hoc trial sorting of neural data on subjective measures can have unintuitive consequences depending on the experimental context (i.e. depending on the criterion that participants adopt). Experimental contexts that induce conservative behavioral responses on subjective measures will overestimate estimates neural correlates of *both* 'unconscious' (unseen) and 'conscious' (seen) conditions, whereas experimental contexts that induce liberal behavioral responding do the reverse. As such, criterion placement threatens the construct validity of neural measures of consciousness.

## Methods
### Simulations
For all simulations, we simulated experiments in Matlab by generating 10,000 normally distributed noise and noise + signal trials using different parameters for the distance between the two distributions as the sensitivity of the system and the criterion that was applied under any given simulation (see OSF https://doi.org/10.17605/OSF.IO/AP23W for all Matlab code and parameter settings). Decoder classification performance was approximated by conceptualizing the difference d between the average signal strength of signal trials and noise trials after post hoc sorting on criterion as a measure of decoder sensitivity under post hoc sorting (ranging from 0 to ∞). This value was converted to AUC classification performance (ranging from 0.5 to 1), using the formula $AUC = \Phi\frac{d'}{\sqrt{2}}$, in which ϕ is the normal cumulative distribution function. The equivalent Matlab code is $AUC_{sorted} = normcdf(d_{sorted}/\sqrt{2})$, for the relevant conversion formula from *d* to AUC see *Ruscio, 2008*.

## Participants and payment

All participants had normal or corrected-to-normal vision and were recruited at the University of Amsterdam (UvA) in partial fulfillment of first year psychology curricular requirements or for monetary reimbursement (€10 per hour). Participants signed a written informed consent form before the start of the experiment, including consent to publish the results from the studies. All procedures were approved by the Ethics Review Board Psychology, section Brain & Cognition of the University of Amsterdam.

### Experiment 1

Sixteen participants (eight females, mean age 24.1 years, SD 1.64, all right-handed) completed three experimental EEG sessions on different days, each session lasting ca. 2 hr. At the beginning of the experiment, participants were informed they could earn a total bonus of €30, on top of their regular pay of €10 per hour or course credit. After completing the last session of the experiment, every participant was paid the full bonus as required by the ethical committee.

### Experiment 2

Thirty-four participants completed three experimental sessions: one behavioral training session lasting approximately 45 min and two experimental EEG session lasting 2.5 hr each. The EEG data from 26 participants were analyzed (14 female, mean age 23.1 years, SD 3.19, 22 right-handed). Four participants were excluded from analysis due having extremely low trial counts on one or more cells after post hoc sorting on the four response levels of the PAS (<5 trials). Four participants were excluded due to equipment failure and/or human error during data collection resulting in corrupted data. Participants had the opportunity to earn an extra €5 in each EEG session, based on the variable payoff scheme described below. All participants were naïve to the purpose of the study.

## Experimental setup and presentation software

Participants completed the experiment in a low-lit, quiet room. Stimuli were shown on a computer monitor, with a refresh rate of 100 Hz. The experiment was created on Presentation software (Neurobehavioral Systems, Inc, Berkeley, CA, https://www.neurobs.com/).

### Experiment 1

Participants were seated approximately 70 cm away from the monitor.

### Experiment 2

Participants rested their heads on a chin rest 73 cm away from the monitor.

## Stimuli and RSVP

Stimuli consisted of a continuous semi-random RSVP of full screen texture patterns (see *Figure 2*). The texture patterns consisted of line elements approximately 0.07° thick and 0.4° long in visual angle. Each texture in the RSVP was oriented in one of four possible directions: 0° [vertical], 45°, 90°, or 135°. After a random intertrial interval (ITI) containing randomly ordered textures, a fixed-order sequence containing 25 textures began. The fifth stimulus of the sequence either contained a texture-defined figure (target, T) or a homogenous texture (non-target, NT). The fixed sequence contained the following orientations: 45°, 90°, 0°, 90°, T/NT, 0°, 90°, 0°, 90°, 0°, 45°, 0°, 135°, 90°, 45°, 0°, 135°, 0°, 45°, 90°, 45°, 90°, 135°, 0°, 135°. This fixed sequence ensured that the visual stimulation surrounding the target was always the same across trials. The non-target was a homogeneous diagonally oriented texture (45° or 135°). The target was the same texture but contained an orientation-defined square in the center, of which the surface elements were orthogonally rotated with respect to the background. Orientation of targets and non-targets was randomly selected, while ensuring that each orientation was used in 50% of trials. The visual angle of the target square was approximately 2.4°. In 75% of trials, a target figure was shown, and in 25% no figure was shown.

### Experiment 1

Target stimuli were presented for 40 ms (i.e. stimulation frequency 25 Hz). The ITI varied randomly between 300 and 2200 ms. The fixation dot was red throughout the experiment.

### Experiment 2

Target stimuli were presented for 30 ms (i.e. stimulation frequency of 33.3 Hz). The ITI varied randomly between 1800 and 2200 ms. The onset of the fixed sequence containing a target or a non-target was signaled by the central fixation dot turning from gray to black. The fixation dot was black during the fixed sequence. After the fixed sequence, the fixation dot changed to green, which indicated that participants could respond.

## Task instructions and payoff induced criterion manipulation

Participants were instructed to detect a target in an RSVP stream by pressing a button, while their criterion was manipulated.

### Experiment 1

Participants were instructed to press a button using their right hand whenever they observed a target in the continuous RSVP. Although the onset of a trial within the continuous stream of textures was not explicitly cued, the similar distribution of reaction times in target and non-target trials suggests that participants used the temporal structure of the task even when no target appeared. See *Kloosterman et al., 2019* for details. In alternating 9-min blocks of trials, we actively biased participants' perceptual decisions by instructing them either to report as many targets as possible while playing an aversive tone after each miss (no button press after presentation of a target, liberal condition), or by instructing them to only report high-certainty targets while playing an aversive tone after each false alarm (button press even when no target was presented, conservative condition). Participants were told their bonus would be diminished by €0.03 after a miss and diminished by €0.10 after a false alarm. Participants were free to respond at any time during a block whenever they detected a target. A trial was considered a target present response when a button press occurred before the fixed-order sequence ended (i.e. within 0.84 s after onset of the fifth texture containing the (non)target, see *Figure 2*). The criterion manipulation switched back and forth after every block, so that each session contained both conservative and liberal criterion blocks.

### Experiment 2

Participants were instructed to determine whether they observed a target in the continuous RSVP using the Perceptual Awareness Scale (PAS; *Ramsøy and Overgaard, 2004*). The PAS is a four-point scale, on which participants rate the strength of their experience of a stimulus from 0 to 3.

The following instructions regarding the PAS were given to participants:

| Response category | Description |
| --- | --- |
| 0 – *No experience* | No impression of a square. |
| 1 – *Brief glimpse* | A feeling that a square was shown. |
| 2 – *Almost clear experience* | Ambiguous experience of a square. A feeling of being almost certain about seeing a square. |
| 3 – *Clear experience* | Non-ambiguous experience of the square. No doubt in one's answer. |

Between blocks, participants were reminded of the description of each response option of the PAS. Participants received additional on-screen feedback if they used one response option for less than 10% of responses in the previous block: '*You are not using all the possible responses on the scale. If this reflects your experience, that is absolutely fine. Otherwise, here are the categories again:*', followed by the PAS descriptions. This was to ensure participants were always aware of using the scale in full, rather than settling into a pattern of choosing between two responses, for example. Participants were further explicitly instructed to respond only according to what they experienced, regardless of the feedback they received during the experiment:

"Even though you receive feedback about the correctness of your responses, it is very important that you keep responding according to what you actually experience, using the Perceptual Awareness Scale. Only press 0 if you are 100% convinced that no square appeared and only press 3 if you are 100% convinced that a square appeared."

Participants were instructed to respond when the fixation dot changed from gray to green, which occurred at the end of the fixed RSVP sequence. Responses given while the fixation date was not green were not recorded. Participants responded using the index finger of their preferred hand, by pressing keys labeled '0', '1', '2', or '3', corresponding to the responses possible on the PAS. As soon as a response was given, the fixation dot changed to gray and the button pressed was displayed in the center of the screen for 60 ms, on top of a stream of textures with a gray fixation dot, so participants could ensure they had pressed the correct key (or, alternatively, correct their finger position for the following trial). An auditory feedback tone was given for either false alarms (conservative condition, responding '1', '2', or '3' when no target was present) or misses (liberal condition, responding '0' when a target was present). In addition, for every tone, €0.01 was deducted from their €5 reward for that session. The criterion manipulation occurred at a session level, so that one feedback scheme was exclusively executed in on session, and the other in the other session. The order of sessions was counterbalanced across participants. There was no break in the stream of textures throughout a block of 144 trials, unless no response was given within the 5 s limit. In this case, participants were shown a screen reading 'Please respond every time the fixation dot is green about what you just experienced.', then given a 5-s countdown before the stream of textures resumed. Throughout the block, the same texture was never repeated twice in a row. No performance feedback was provided at the end of a block. Participants were informed of how much of the extra reward they earned in each session at the end of all sessions.

## EEG sessions
### Experiment 1
Prior to EEG recording in the first session, participants performed a 10-min practice run of both conditions, in which visual feedback directly after a miss (liberal condition) or false alarm (conservative) informed participants about their mistake, allowing them to adjust their decision bias accordingly. During EEG recording, participants performed six blocks per session lasting ca. 9 min each. During a block, participants continuously monitored the screen and were free to respond by button press whenever they thought they saw a target. Each block contained 240 trials, of which 180 target and 60 non-target trials. The condition of the first block of a session was counterbalanced across participants. There were short breaks between blocks, in which participants indicated when they were ready to begin the next block.

### Experiment 2
Prior to collecting EEG, each subject underwent a practice session. The practice session started with a slower version of the task, so participants could familiarize themselves with the structure of the trials, and clearly identify the target, and in which they were familiarized with the PAS scale. Only participants that were able to perform the task with a reasonable accuracy of 30% (hit rate minus false alarm rate) were invited for the subsequent EEG sessions. EEG was collected in two different experimental sessions. A session contained ten blocks of either the liberal or conservative condition (counterbalanced across participants). Each block contained 144 trials (108 target trials, 36 non-target trials), and lasted approximately 9 min.

## EEG recording
Continuous EEG data were recorded at 512 HZ using a 64-channel BioSemi Active-Two system (BioSemi, Amsterdam, The Netherlands). Two external electrodes were placed on the earlobes, to be used as a reference. Electroocculargraphy (EOG) was recorded using four electrodes: on the outer side of each eye (horizontal) and above and below the left eye (vertical). Horizontal and vertical EOG electrodes were referenced against each other, to obtain information about horizontal eye movements, and vertical eye movements and blinks, respectively. Triggers were sent at the time of response and target presentation, recording the orientation and type of trial (target or catch).

### Experiment 1

EEG was recorded from a 48-electrode EEG cap that was slightly modified to include I1 and I2 next to Iz. Other electrodes were placed according to the 10–20 system. The complete list of electrodes was AF3, AF4, C3, C4, CP1, CP2, CP3, CP4, CP5, CP6, Cz, F3, F4, F7, F8, FC1, FC2, FC5, FC6, Fp1, Fp2, Fz, I1, I2, Iz, O1, O2, Oz, P1, P10, P2, P3, P4, P5, P6, P7, P8, P9, PO3, PO4, PO7, PO8, POz, Pz, T7, T8, TP7, TP8.

### Experiment 2

EEG was recorded from a standard 64-electrode EEG cap, according to the 10–20 system. The complete list of electrodes was AF3, AF4, AF7, AF8, AFz, C1, C2, C3, C4, C5, C6, CP1, CP2, CP3, CP4, CP5, CP6, CPz, Cz, F1, F2, F3, F4, F5, F6, F7, F8, FC1, FC2, FC3, FC4, FC5, FC6, FCz, FT7, FT8, Fp1, Fp2, Fpz, Fz, Iz, O1, O2, Oz, P1, P10, P2, P3, P4, P5, P6, P7, P8, P9, PO3, PO4, PO7, PO8, POz, Pz, T7, T8, TP7, TP8.

## EEG pre-processing

The EEG data from Experiment 1 were pre-processed as described in detail in *Kloosterman et al., 2019*, with the only exception that for these analyses no detrending was applied to the data. The data from Experiment 2 were pre-processed using a very similar pre-processing pipeline, as described next. All pre-processing and subsequent analyses were conducted using EEGLAB (*Delorme and Makeig, 2004*), FieldTrip (*Oostenveld et al., 2011*), and/or the ADAM toolbox (*Fahrenfort et al., 2018*) using MATLAB code. All data were referenced to the average voltage of two electrodes attached to the earlobes. Channel locations were looked up according to the standard 10–5 BESA cap. Data were downsampled to 256 Hz to reduce time and space required for further pre-processing and analysis. The continuous EEG data were epoched between 100 ms before target presentation and 750 ms after target presentation. An independent component analysis (ICA) was used on epoched and demeaned data to identify and remove eye-blinks. Finally, the data were transformed to scalp current density (CSD) using spherical splines (*Perrin et al., 1989*), after which the data were baseline-corrected using the interval (–100, 0) ms prior to decoding. No high- or low-pass filtering was applied to the data to preclude temporal displacements (*van Driel et al., 2021*; *Vanrullen, 2011*).

## EEG decoding analyses

Trials were balanced such that the number of trials within each stimulus–response combination were always the same between the liberal and the conservative condition. To achieve this, trials were randomly selected from the stimulus–response condition with the overrepresented stimulus class to match the condition with fewer responses in number. This was done to ensure that differences between the liberal and conservative conditions could not arise just because more trials were included in one condition than in the other.

EEG data were analyzed using the ADAM toolbox (*Fahrenfort et al., 2018*), a MATLAB toolbox for multivariate pattern analysis of EEG data. The train–test procedure used to classify EEG data was a leave-one-person-out cross validated decoding scheme. In this procedure, all sessions from each participant were merged, and electrodes were used as features to train an LDA classifier to discriminate between targets (figures) and no targets (homogenous textures) for every sample in the epoch (–100, 750) ms.

Next, these classifiers were used to test the datasets from all participants, except the one that the classifier had been trained on. This procedure was repeated until all classifiers had tested all datasets, except data from the same participant. For Experiment 1 (*N* = 16), this resulted in 16*15 = 240 classifier performance sets, and for Experiment 2 (*N* = 26), this resulted in 26*25 = 650 classifier performance sets. Classifier performance sets that were tested on the same person were subsequently averaged, returning to 16 classifier performance sets for Experiment 1, and 26 classifier performance sets for Experiment 2, which were used for subsequent group level statistics. A graphical depiction of this leave-one-person-out cross validation procedure is shown in *Figure 3— figure supplement 2*. The procedure ensured that train and test sets were fully independent, while maximizing the available training data to evaluate test data, as well as maximizing the generalizability of the results within the tested population. The accuracy measure to establish classifier performance was AUC. Classifiers were trained and tested using the occipitoparietal electrodes

in both datasets: Iz, O1, O2, Oz, P1, P10, P2, P3, P4, P5, P6, P7, P8, P9, PO10, PO3, PO4, PO7, PO8, PO9, POz, Pz. Group level ANOVAs on classifier performance scores were performed in JASP (*JASP Team, 2023*).

## Additional information

### Competing interests

Simon van Gaal: Reviewing editor, *eLife*. The other authors declare that no competing interests exist.

### Funding

| Funder | Grant reference number | Author |
|---|---|---|
| HORIZON EUROPE European Research Council | 10.3030/715605 | Simon van Gaal |

The funders had no role in study design, data collection, and interpretation, or the decision to submit the work for publication.

### Author contributions

Johannes Jacobus Fahrenfort, Conceptualization, Resources, Data curation, Software, Formal analysis, Supervision, Validation, Visualization, Methodology, Writing – original draft, Project administration, Writing – review and editing; Philippa A Johnson, Niels A Kloosterman, Data curation, Investigation, Project administration, Writing – review and editing; Timo Stein, Writing – review and editing; Simon van Gaal, Supervision, Funding acquisition, Writing – review and editing

### Author ORCIDs

Johannes Jacobus Fahrenfort ![ORCID] https://orcid.org/0000-0002-9025-3436
Philippa A Johnson ![ORCID] https://orcid.org/0000-0002-6125-3138
Niels A Kloosterman ![ORCID] https://orcid.org/0000-0002-1134-7996
Timo Stein ![ORCID] https://orcid.org/0000-0002-8484-0933
Simon van Gaal ![ORCID] https://orcid.org/0000-0001-6628-4534

### Ethics

Participants signed a written informed consent form before the start of the experiment, including consent to publish the results from the studies. All procedures were approved by the Ethics Review Board Psychology, section Brain & Cognition of the University of Amsterdam.

Reviewer #2 (Public review): https://doi.org/10.7554/eLife.102335.4.sa1
Author response https://doi.org/10.7554/eLife.102335.4.sa2

## Additional files

### Supplementary files

Supplementary file 1. Tables containing the full ANOVA results of all reported statistical tests. (**a**) Repeated measures ANOVA after post hoc sorting. In cases where Mauchly's test of sphericity was violated, the Greenhouse–Geisser corrected values are provided below it on the second row. (**b**) Repeated measures ANOVA control (no post hoc sorting). In cases where Mauchly's test of sphericity was violated, the Greenhouse–Geisser corrected values are provided below it on the second row. (**c**) Repeated measures ANOVA for Experiment 1 (detection), after post hoc sorting. There were no violations of Mauchly's test of sphericity. (**d**) Repeated measures ANOVA for Experiment 2 (PAS), after post hoc sorting. In cases where Mauchly's test of sphericity was violated, the Greenhouse–Geisser corrected values are provided below it on the second row.

MDAR checklist

### Data availability

All data and scripts have been deposited on the Open Science Framework.

The following previously published dataset was used:

| Author(s) | Year | Dataset title | Dataset URL | Database and Identifier |
| --- | --- | --- | --- | --- |
| Fahrenfort JJ, Johnson P, Kloosterman N, Stein T, van Gaal S | 2024 | Criterion placement threatens the construct validity of neural measures of consciousness | https://doi.org/10.17605/OSF.IO/AP23W | Open Science Framework, 10.17605/OSF.IO/AP23W |

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
