## [Editor Report · eLife Assessment]

This **fundamental** study provides a critical challenge to a great many studies of the neural correlates of consciousness that were based on post hoc sorting of reported awareness experience. The evidence supporting this criticism is **compelling**, based on simulations and decoding analysis of EEG data. The results will be of interest not only to psychologists and neuroscientists but also to philosophers who work on addressing mind-body relationships.

---

## [Referee Report · Reviewer #2 (Public review)]

Summary:

The study investigates the potential influence of the response criterion on neural decoding accuracy in consciousness and unconsciousness, utilizing either simulated data or reanalyzing experimental data with post-hoc sorting data.

Strengths:

When comparing the neural decoding performance of Target versus NonTarget with or without post-hoc sorting based on subject reports, it is evident that response criterion can influence the results. This was observed in simulated data as well as in two experiments that manipulated the subject response criterion to be either more liberal or more conservative. One experiment involved a two-level response (seen vs unseen), while the other included a more detailed four-level response (ranging from 0 for no experience to 3 for a clear experience). The findings consistently indicated that adopting a more conservative response criterion could enhance neural decoding performance, whether in conscious or unconscious states, depending on the sensitivity or overall response threshold.

The uneven distribution of trails for Target (75%) and NonTarget (25%) was identified as a potential weakness in the initial review of this study. Nevertheless, we support the authors' assertion that their analysis methodology validates comparing liberal and conservative approaches. Future investigations could further explore differences between liberal and conservative on different ratios of Target vs NonTarget, particularly when the proportion of Target matches or falls below that of NonTarget.

---

## [Author Response]

The following is the authors’ response to the previous reviews

**Reviewer #1 (Public review):**
Summary:The study aimed to investigate the significant impact of criterion placement on the validity of neural measures of consciousness, examining how different standards for classifying a stimulus as 'seen' or 'unseen' can influence the interpretation of neural data. They conducted simulations and EEG experiments to demonstrate that the Perceptual Awareness Scale, a widely used tool in consciousness research, may not effectively mitigate criterion-related confounds, suggesting that even with the PAS, neural measures can be compromised by how criteria are set. Their study challenged existing paradigms by showing that the construct validity of neural measures of conscious and unconscious processing is threatened by criterion placement, and they provided practical recommendations for improving experimental designs in the field. The authors' work contributes to a deeper understanding of the nature of conscious and unconscious processing and addresses methodological concerns by exploring the pervasive influence of criterion placement on neural measures of consciousness and discussing alternative paradigms that might offer solutions to the criterion problem.

The study effectively demonstrates that the placement of criteria for determining whether a stimulus is 'seen' or 'unseen' significantly impacts the validity of neural measures of consciousness. The authors found that conservative criteria tend to inflate effect sizes, while liberal criteria reduce them, leading to potentially misleading conclusions about conscious and unconscious processing. The authors employed robust simulations and EEG experiments to demonstrate the effects of criterion placement, ensuring that the findings are well-supported by empirical evidence. The results from both experiments confirm the predicted confounding effects of criterion placement on neural measures of unconscious and conscious processing.

The results are consistent with their hypotheses and contribute meaningfully to the field of consciousness research.

We would like to thank reviewer 1 for their positive words and for taking the time to evaluate our manuscript.

**Reviewer #2 (Public review):**
Summary:The study investigates the potential influence of the response criterion on neural decoding accuracy in consciousness and unconsciousness, utilizing either simulated data or reanalyzing experimental data with post-hoc sorting data.Strengths:When comparing the neural decoding performance of Target versus NonTarget with or without post-hoc sorting based on subject reports, it is evident that response criterion can influence the results. This was observed in simulated data as well as in two experiments that manipulated subject response criterion to be either more liberal or more conservative. One experiment involved a two-level response (seen vs unseen), while the other included a more detailed four-level response (ranging from 0 for no experience to 3 for a clear experience). The findings consistently indicated that adopting a more conservative response criterion could enhance neural decoding performance, whether in conscious or unconscious states, depending on the sensitivity or overall response threshold.Weaknesses:(1) In the realm of research methodology, conducting post-hoc sorting based on subject reports raises an issue. This operation leads to an imbalance in the number of trials between the two conditions (Target and NonTarget) during the decoding process. Such trial number disparity introduces bias during decoding, likely contributing to fluctuations in neural decoding performance. This potential confounding factor significantly impacts the interpretation of research findings. The trial number imbalance may cause models to exhibit a bias towards the category with more trials during the learning process, leading to misjudgments of neural signal differences between the two conditions and failing to accurately reflect the distinctions in brain neural activity between target and non-target states. Therefore, it is recommended that the authors extensively discuss this confounding factor in their paper. They should analyze in detail how this factor could influence the interpretation of results, such as potentially exaggerating or diminishing certain effects, and whether measures are necessary to correct the bias induced by this imbalance to ensure the reliability and validity of the research conclusions.

We would like to thank reviewer 2 for their positive words and for taking the time to evaluate our manuscript. In response to this asserted weakness, we would like to point out that the issue of trial imbalances was already comprehensively addressed in the manuscript. No trial imbalances are present in the analyzed data for any of the conditions, so that none of our reported results could have been impacted by this. This was done through the following set of measures:

(1) Training data (method section): “a linear discriminant analytic (LDA) classifier was trained for each participant using all trials from all sessions (3 sessions in Experiment 1, 2 sessions in Experiment 2) to discriminate target from no-target trials based on EEG data, irrespective of seen/unseen responses and irrespective of the response criterion. To maximize signal-to-noise ratio, we applied a leave-one-person-out cross validated decoding scheme by using all classifiers from all participants except the participants that was being tested (separately for Experiment 1 and for Experiment 2). This leave-one-person-outcross validation procedure maximized the available data for training without requiring k-foldingon subsets of cells with low response counts, so that all test sets were classified by the same fully independent classifiers. A single time series of classification performance across time was obtained for every participant (every testing set) by averaging classification performance across all classifiers that tested that set (see Methods and supplementary Figure S2 for details).”

This leave-one-person-outcross validation scheme made surre that no trial selection needed to be performed to analyze conservative or liberal conditions. Both conditions were classified using the same classifier, consisting of all data from the other participants.

(2) Testing data (methods section): “To ensure that differences resulting from post hoc sorting could not be explained by differences in signal-to-noise ratio resulting from disparities in trial counts in the testing set, we equated trial counts between the liberal and conservative condition within each participant by randomly selecting the same number of trials from overrepresented cells (for Experiment 1, this was done at the level of ‘seen’ and ‘unseen’ responses, for experiment 2 the trial counts were equated at eachof the PAS levels, see methods for details). As a result, response-contingent conditions in the liberal and conservative conditions had identical input for all classification analyses. Although different trial counts in the testing set might affect the precision with which AUC is estimated in a decoding analysis, it does not affect the size of AUC itself. Trial count equation was merely performed tomake sure the liberal and conservative condition were as comparable as possible.”

Indeed, we also report at the end of this section that running the same analyses without selecting trials in the test set yielded qualitatively identical results: “Analyzing the data without equating trial counts resulted in qualitatively identical results.”

To remove any lack of clarity about this, we now also briefly report in the beginning of the discussion section that the results cannot be explained by unequal trial counts:

“We found that in both experiments, criterion shifts modulated effect size in neural measures of ‘unconscious’ (unseen) and/or ‘conscious’ (seen) processing, and that this happens even though the conservative and liberal condition used the same independent training data (identical classifiers), and even though the trial counts in the test sets were equated for the conservative and liberal condition.”

**Reviewer #3 (Public review):**
Summary:Fahrenfort et al. investigate how liberal or conservative criterion placement in a detection task affects the construct validity of neural measures of unconscious cognition and conscious processing. Participants identified instances of "seen" or "unseen" in a detection task, a method known as post hoc sorting. Simulation data convincingly demonstrate that, counterintuitively, a conservative criterion inflates effect sizes of neural measures compared to a liberal criterion. While the impact of criterion shifts on effect size is suggested by signal detection theory, this study is the first to address this explicitly within the consciousness literature. Decoding analysis of data from two EEG experiments further shows that different criteria lead to differential effects on classifier performance in post hoc sorting. The findings underscore the pervasive influence of experimental design and participant reports on neural measures of consciousness, revealing that criterion placement poses a critical challenge for researchers.Strengths and WeaknessesOne of the strengths of this study is the inclusion of the Perceptual Awareness Scale (PAS), which allows participants to provide more nuanced responses regarding their perceptual experiences. This approach ensures that responses at the lowest awareness level (selection 0) are made only when trials are genuinely unseen. This methodological choice is important as it helps prevent the overestimation of unconscious processing, enhancing the validity of the findings.The authors also do a commendable job in the discussion by addressing alternative paradigms, such as wagering paradigms, as a possible remedy to the criterion problem (Peters & Lau, 2015; Dienes & Seth, 2010). Their consideration of these alternatives provides a balanced view and strengthens the overall discussion.Our initial review identified a lack of measures of variance as one potential weakness of this work. However we agree with the authors' response that plotting individual datapoints for each condition is indeed a good visualization of variance within a dataset.Impact of the Work:This study effectively demonstrates a phenomenon that, while understood within the context of signal detection theory, has been largely unexplored within the consciousness literature. Subjective measures may not reliably capture the construct they aim to measure due to criterion confounds. Future research on neural measures of consciousness should account for this issue, and no-report measures may be necessary until the criterion problem is resolved.

We thank reviewer 3 for their positive words and for taking the time to evaluate our manuscript.